# The Use of Non-Nutritive and Low-Calorie Sweeteners in 19,915 Local and Imported Pre-Packaged Foods in Hong Kong

**DOI:** 10.3390/nu13061861

**Published:** 2021-05-29

**Authors:** Billy Yin Sing O, Daisy H. Coyle, Elizabeth K. Dunford, Jason H. Y. Wu, Jimmy Chun Yu Louie

**Affiliations:** 1School of Biological Sciences, Faculty of Science, The University of Hong Kong, Hong Kong 999077, China; billioys@connect.hku.hk; 2Food Policy Division, The George Institute for Global Health, University of New South Wales, Kensington, NSW 2052, Australia; dcoyle@georgeinstitute.org.au (D.H.C.); edunford@georgeinstitute.org.au (E.K.D.); jwu1@georgeinstitute.org.au (J.H.Y.W.); 3Department of Nutrition, Gillings Global School of Public Health, The University of North Carolina at Chapel Hill, Chapel Hill, NC 27599, USA

**Keywords:** non-nutritive sweeteners, low-calorie sweeteners, intense sweeteners, sugar substitutes, pre-packaged foods, Hong Kong

## Abstract

This study aims to examine the use of non-nutritive (NNSs) and low-calorie sweeteners (LCSs) in pre-packaged foods in Hong Kong and the differences in the number of NNSs/LCSs used between products from different regions. In a cross-sectional audit, the types of NNSs/LCSs used in 19,915 pre-packaged foods in Hong Kong were examined by searching the ingredients list of the included products for keywords related to 20 common NNSs/LCSs and their respective E-numbers. Prevalence of use of NNSs and LCSs, the co-presence of NNSs/LCSs and free sugar ingredients (FSI), and the number of NNSs/LCSs used in the included foods were computed. Pearson’s *χ*^2^ test was used to compare the total number of NNSs and/or LCSs used in food items from different regions. Sucralose (E955) was the most commonly used NNS (1.9%), followed by acesulfame K (E950, 1.6%). Sorbitol was the most commonly used LCS (2.9%). Overall, the use of LCSs was less common compared with NNSs (3.7% vs. 4.5%). The use of different types of NNSs varied substantially between food types. Notably, 20.2% of potato crisps and 15.2% of other crisps or extruded snacks contained at least one NNS and/or LCS. Co-presence of FSIs and NNSs/LCSs were most common in confectionery (15.7%) and snack foods (15.5%). Asian prepackaged foods were more likely to contain NNSs/LCSs (10.1%) compared with those from other regions. To conclude, NNSs/LCSs were used in a wide range of non-diet pre-packaged products which could be a public health concern due to their higher consumption frequencies than “diet” products.

## 1. Introduction

Excessive free sugar consumption has been proposed to be a major driver of the worldwide obesity epidemic [1,2]. Public health agencies and healthcare professionals have long been advocating for a reduction in free sugar content in processed foods, as the majority of the population’s daily free sugar intake comes from these items [3]. The latest guidelines from the World Health Organization (WHO) recommend individuals limit their intake of free sugars (i.e., monosaccharides and disaccharides added to foods by manufacturers, including sugars that are naturally present in honey, syrups and fruit juices) to no more than 5–10% of total energy intake [4].

Reformulation of packaged foods to reduce the sugar content is a public health priority globally. Many governments worldwide have imposed a sugar tax (e.g., the UK, France, Mexico) and/or set voluntary reformulation targets for sugar [5,6,7,8]. In response to these sugar reduction policies as well as consumer demand for lower sugar foods, there has been an increase in the use of non-nutritive sweeteners (NNSs, e.g., aspartame) and low-calorie sweeteners (LCSs, e.g., xylitol) in recent years [4,9,10,11,12,13]. Due to their higher sweetness intensity, these sweeteners can provide a similar level of sweetness as sugar but with substantially fewer calories, and are, therefore, a popular alternative to sugar for manufacturers. A recent audit of packaged foods sold in Australia, Canada, the USA and Mexico reported that NNSs were present in a wide variety of food and beverages, such as confectionery, dairy products, desserts, soft drinks and juice [14]. Co-presence of NNSs/LCSs with free sugar has also become more common [11]. This could be a potential public health concern considering nutrition labels are not required to display the amount of NNSs/LCSs used [15,16,17], and therefore, consumers may be unaware that some sugars in their favorite foods/beverages are being replaced by NNSs/LCSs [18].

While regulatory agencies around the world have deemed the use of NNSs/LCSs to be safe at the dosage commonly found in food products [19,20,21], the impact of NNSs/LCSs intake on health outcomes remains controversial [22,23,24,25,26], with some studies suggesting that the use of NNSs/LCSs may be associated with obesity and chronic diseases such as diabetes or cardiovascular diseases [23,24]. Previous studies and meta-analyses [27,28,29], however, have often considered NNSs/LCSs as a single entity which have not allowed the attribution of the observed associations with health outcomes to a particular type of NNS/LCS. It has been suggested that NNSs/LCSs may have different effects on health outcomes [30], e.g., weight loss [31]; generalizing the potential health effects of a particular type of NNSs/LCSs to other NNSs/LCSs is likely inappropriate [32]. Given there are more than 20 types of NNSs/LCSs commonly used in pre-packaged foods, it is important to examine the prevalence of use of individual types of NNSs/LCSs in the food supply to inform risk assessment and policy development.

To date, there has been no research that has examined the use of NNSs/LCSs in different types of packaged foods in Hong Kong. Given around 90% of the Hong Kong food supply is imported from various parts of the world [33], and the increasingly common use of NNSs/LCSs in other countries as described above, the aim of this study was to examine the use of NNSs/LCSs in pre-packaged foods in Hong Kong and the differences in the number of NNSs/LCSs used between products from different regions.

## 2. Materials and Methods

### 2.1. Data Source

This cross-sectional examination of the packaged food supply in Hong Kong used data from the 2019 FoodSwitch Hong Kong monitoring and surveillance database. This database is the only comprehensive pre-packaged food database with nutrient and ingredient information available in Hong Kong [34]. The largest megastore of each of the following major chain supermarkets/retailers in Hong Kong was visited due to the wider range of products available: Park’ n Shop and Wellcome (both selling a wide range of local and imported groceries including those from typical household brands), AEON (selling a wide range of popular Japanese imported products), City’Super (selling a wide range of niche imported products from all over the world) and Marks & Spencer (selling a wide range of its home brand products originating from the UK). The stores were located in different geographical areas, but all were affluent areas [34]. Together, these retailers account for more than 70% of the market share of pre-packaged foods in Hong Kong [35], while the remaining 30% are small retailers, internet stores and others. Data in the monitoring and surveillance database were collected by trained research assistants, who visited these stores and took several photographs of pre-packaged products that displayed a nutrition information panel. Photographs were taken of the barcode, front-of-package, nutrition information panel and the ingredients list using a bespoke smartphone application [36]. While FoodSwitch Hong Kong also maintains a crowd-sourced database, it was not used in the current study.

### 2.2. Data Entry and Processing

For each food item, the barcode, name and brand of the product, ingredients, nutrition information and country of origin were recorded into the FoodSwitch database. The countries of origin were identified using the 3-digit prefix of the Global Trade Item Number (GTIN) standard barcode [37], which identifies the issuance country, as a proxy if that corresponds to a single country, e.g., “489” for Hong Kong. For items which have GTIN prefixes that correspond to more than one issuance country, e.g., “540” to “549” corresponding to Belgium and Luxemburg, or items with non-GTIN-standard barcodes, the on-pack declaration of country of origin was used. The countries of origin were then grouped into five regions, namely Asia, Europe, North America, Oceania and others (including South America, the Middle East and Africa). These food items were then categorized into 18 major food groups (and their respective minor food groups) according to the food categorization system developed by The Global Food Monitoring Group [38].

### 2.3. Exclusion Criteria

Several food groups, namely alcohols (*n* = 27), special foods (*n* = 599), vitamins and supplements (*n* = 5) and foods unable to be categorized (*n* = 41) were excluded from analysis, leaving 14 major food groups. These categories were excluded as they are not required to carry the standard nutrition information panel of Hong Kong, and/or are not expected to be a major dietary contributor of NNSs/LCSs. Of the remaining 20,450 items, 420 were excluded due to being a duplicate product (i.e., same item in different pack sizes), 113 items were excluded for not having an ingredients list and two items were excluded for having multiple nutrition information panels. This left 19,915 items for the main analysis.

### 2.4. Identification of NNSs/LCSs and Free Sugar Ingredients (FSIs)

In this study, a total of 20 different types of NNSs and LCSs were examined. The NNSs examined included acesulfame potassium (acesulfame K, E950), advantame (E969), alitame (E956), aspartame (E951), aspartame-acesulfame salt (E962), cyclamic acid (E952), saccharin (E954), sucralose (E955), thaumatin (E957), neotame (E961), steviol glycosides (E960), and neohesperidin dihydrochalcone (E959). The LCSs examined included luo han guo (monk fruit), erythritol, isomalt, lactitol, maltitol, mannitol, sorbitol and xylitol. All NNSs except advantame and neohesperidin dihydrochalcone are permitted for use in Hong Kong [39], and there are no specific restrictions on LCSs in Hong Kong. The official and alternative names of these sweeteners and their respective E-codes were used as keywords for their identification in the ingredients lists. Individual types of free sugar ingredients (FSIs) were identified according to the method described by Bernstein et al. [40]. The food items were then coded as “no FSI” or “with FSI” accordingly.

### 2.5. Data Cleaning and Statistical Analysis

Data were analyzed using SPSS (version 25.0, IBM Corp. Ltd., New York, NY, USA). Descriptive statistics were computed for the prevalence of use of NNSs and LCSs, the co-presence of NNSs/LCSs and FSI, and the number of NNSs/LCSs used in the included food items, stratified by food category and region of origin where appropriate. Pearson’s *χ*^2^ test was used to compare the total number of NNSs and/or LCSs used in food items from different regions. As this was an exploratory study, a two-tailed *p* < 0.05 was considered statistically significant.

## 3. Results

### 3.1. Types of NNSs and LCSs Used

Overall, 4.5% of the included items contained at least one NNS (Appendix A). Of all the NNSs examined, sucralose was the most commonly used, which was found in 1.9% of all items, followed by acesulfame K (1.6%), aspartame (1.3%) and stevia (1.0%) (Figure 1). Alitame and thaumatin were not used in the included products, and the use of the other types of NNSs was infrequent. Neohesperidin dihydrochalcone and advantame were found in one soft drink and one noodle product, respectively (Appendix A). Figure 2 shows the prevalence of use of different LCSs by food categories. Sorbitol was the most commonly used LCS, which was found in 2.9% of the included products, followed by maltitol (0.6%). Overall, the use of LCSs was less common compared with NNSs (3.7% vs. 4.5%; Appendix A).

### 3.2. Use of NNSs/LCSs by Food Category

The use of the different types of NNSs and LCSs varied substantially between food types. For example, aspartame was more commonly used in snack foods (6.0%), whereas acesulfame K was more commonly used in confectionery items (8.5%) (Figure 1). LCSs were more commonly found in confectionery (17.8%) and cereal and grain products (16.5%) (Figure 2). Notably, savory items such as 20.2% of potato crisps and 15.2% of other crisps or extruded snacks available in Hong Kong were found to contain at least one NNS and/or LCS (Figure 3). When examining the number of NNSs and/or LCSs used (Figure 4), a large proportion of NNSs/LCSs-containing confectionery, non-alcoholic beverages and snack foods utilized two or more NNSs and/or LCSs, while the majority of NNSs/LCSs-containing items in most other categories used only one NNS or LCS.

### 3.3. Co-Presence of NNSs, LCSs and FSIs

The prevalence of co-presence of NNSs, LCSs and FSIs is summarized in Table 1. The majority of items (57.7%) contained FSIs only, followed by items with no FSIs, NNSs nor LCSs (35.1%). Overall, 5.9% of items contained FSIs together with either NNSs or LCSs or both. Only 1.4% of items contained NNSs and/or LCSs without FSIs. Co-presence of FSIs and NNSs/LCSs were more common in confectionery (15.7%), snack foods (15.5%) and non-alcoholic beverages (7.5%). The non-alcoholic beverages category had the highest proportion of items using NNSs and/or LCSs without FSIs (2.9%).

### 3.4. Use of NNSs/LCSs by Region of Origin

When examined by region of origin, a higher proportion of items from Asia used NNSs and/or LCSs (10.1%), followed by North America (4.6%) (Figure 5). Notably, use of NNSs and/or LCSs was more common in snacks (25.5%), bread and bakery products (12.3%), convenience foods (7.7%), fruits and vegetables (8.3%) (7.4%), fish and fish products and cereal and grain products (7.0%) from Asia compared with those from other regions. On the other hand, a substantially higher proportion (25.9%) of confectionery items from North America utilized two or more NNSs/LCSs than confectionery from the other regions. Products from Europe and Oceania infrequently utilize NNSs and or LCSs, and NNSs and LCSs are mostly found in non-alcoholic beverages and bread and bakery products. None of the products from South America, the Middle East and Africa contained any NNSs/LCSs.

## 4. Discussion

This study found that sucralose, acesulfame K, aspartame and stevia were the most commonly used NNSs, while sorbitol was the most commonly used LCS, which is consistent with previous research in other countries [11]. The types of NNSs/LCSs used differed between food categories, possibly due to the heat and chemical stability of the NNSs/LCSs under different conditions. For example, aspartame was more commonly found in products that do not undergo heat treatment, as it is known to breakdown at high temperatures [41]. In contrast, heat-stable NNSs such as sucralose were more commonly used in heat-treated products such as bakery items. Nonetheless, our cross-sectional audit showed a higher proportion of foods contained stevia and sucralose compared with audits from other countries [11,42]. Although comprehensive literature reviews conducted by regulatory agencies [19,20,21] have concluded that concerns regarding “older” sweeteners such as aspartame were not supported by scientific data, manufacturers may opt to replace these controversial sweeteners with those not as controversial such as stevia and/or sucralose in response to consumers’ preference and demand [43]. Newer sweeteners entering the market as well as the ongoing controversies regarding the safety of NNSs/LCSs will also affect food company decisions on the type of NNSs/LCSs to use. This highlights the importance of regular monitoring and surveillance on the use of NNSs/LCSs to allow examination of change in their use over time.

“Diet” products generally use NNSs/LCSs to replace sugars to achieve a lower calorie content. However, an important finding of our study was that a sizable proportion of ordinary non-diet products contained at least one NNS/LCS. Indeed, more than 15% of snack foods (mostly savory) included in this study contained at least one NNS, which is in opposition with the commonly expected usage of NNSs/LCSs, i.e., for replacement of free sugars in sweet “diet” products. This may have a negative public health impact. Studies have suggested that even young adults, who are usually more health-conscious/literate than other age groups, may not realize that they have consumed NNSs/LCSs [18,44]. Furthermore, patients with phenylketonuria (PKU) need to avoid products containing aspartame. Although labelling regulations around the world require an on-pack warning regarding the presence of aspartame [45,46], this is not mandatory in Hong Kong [47]. The presence of aspartame in non-diet products may mean that PKU patients will need to be further reminded about the possibility of such unexpected exposure, and the importance of checking the declaration on packaging and ingredients list for all packaged products.

We also observed a significant inter-regional difference in the number of NNSs/LCSs used, which may be reflective of the regulatory requirements of the respective countries as well as the taste preference of consumers in the region. For example, the higher use of NNSs amongst Asian snacks compared with those from other regions may be because Asian potato chips tend to have a sweeter taste profile (e.g., honey soy flavor) compared with Western potato chips (e.g., cheese flavor), while the substantially higher use of NNSs/LCSs in North American confectionery may reflect the preference of stronger sweet taste of North American consumers. In contrast, the lower use of NNSs/LCSs in packaged foods in Europe and Oceania may be due to poorer consumer acceptance of NNSs/LCSs [48]. For Hong Kong, where the majority of packaged foods are imported [33], this poses a unique challenge to government agencies (in terms of regulation) and health professionals (in terms of nutrition education). These findings highlight that NNSs/LCSs are no longer just prevalent in “diet” products. As such, nutrition education should focus on teaching consumers how to identify the presence of NNSs/LCSs, particularly as they appear to be found in a wide range of pre-packaged foods.

Another interesting finding from our study is that the co-use of NNSs/LCSs with FSIs was more common compared with exclusive use of NNSs and/or LCSs, which echoes the finding of a recent US study [11]. In these products, rather than a total replacement of free sugars with NNSs/LCSs, manufacturers commonly reduce the free sugar content and use NNS/LCS to compensate for this loss of sweetness to ensure the product has a similar taste profile to the original product. Increased use of NNSs/LCSs is likely due to the recent emergence of sugar reduction strategies set by governments around the world, including sugar reformulation targets, portion size guidelines and taxation of sugar-sweetened beverages [5,6,7,8].

Although the replacement of free sugars with NNSs and/or LCSs could result in an immediate reduction in free sugars and caloric intake, the potential health consequences of regular consumption of NNSs/LCSs remain a hotly debated topic [49,50,51]. The co-occurrence of NNSs/LCSs and FSI as observed in this study may also carry potential risks to metabolic and physiological disruption [52,53]. For example, the sweet sensation elicited by sugars normally leads to an anticipation of the arrival of food in the gastrointestinal tract [53], and the co-presence of FSI with NNSs/LCSs may result in a mismatch in the “anticipated calorie” and actual calorie consumed, which may lead to caloric compensation in subsequent meal [54].

The strengths of this study include the use of a large dataset of commonly available pre-packaged foods in Hong Kong manufactured in various regions of the world that were collected using a standard, systematic protocol. Moreover, it is the first comprehensive audit of this kind in the Asian region. We have also performed a comprehensive search of NNSs/LCSs/FSIs keywords which covered most, if not all, of the NNSs/LCSs/FSIs that are permitted for use in food products, and we also examined the difference of their use by their region of origin. Furthermore, we assessed the prevalence of use of individual types of NNSs/LCSs to better inform risk assessment as research has shown that the health effects of individual types of NNSs/LCSs do differ [32].

However, we alert the readers to several limitations. First, the dataset contains a disproportionately high number of snacks compared with similar studies from other countries [11,14], which may have biased the overall prevalence of use of NNSs/LCSs. Nonetheless, this is reflective of the local food supply in Hong Kong, and the subgroup analyses provided details on the use of NNSs/LCSs in different food categories. Second, we relied on the ingredients information to identify the presence of NNSs/LCSs without confirming the actual amount used by laboratory analysis. However, doing so is costly, time-consuming and financially unfeasible. This also means we were unable to conduct a full risk assessment of these unexpected exposures of NNSs/LCSs in non-diet products, and we cannot confirm how the findings translate to actual purchases or intake. Future studies would benefit from linking data on the prevalence of use with purchasing or intake data to understand what the findings mean in terms of what consumers are actually eating. Given the high prevalence of use of NNSs/LCSs in non-diet products, future risk assessments in Hong Kong and possibly around the world examining the risk of NNSs/LCSs exposure should include these products in the modelling.

## 5. Conclusions

NNSs/LCSs were used in a wide range of pre-packaged foods in Hong Kong, the majority of which were used in non-diet products such as savory snacks. This could be a significant public health concern, as consumers may be unaware of their presence. Where technically and organoleptically possible, manufacturers are encouraged to use other reformulation strategies than replacing free sugars with NNSs/LCSs. Future risk assessments should consider the contribution of non-diet products to NNSs/LCSs exposure, and regulations regarding NNSs/LCSs usage should be updated to reflect the new risk profiles. Consumer education programs to increase the awareness of NNSs/LCSs usage in pre-packaged foods, as well as their potential health implications, are also needed.

## Figures and Tables

**Figure 1 nutrients-13-01861-f001:**
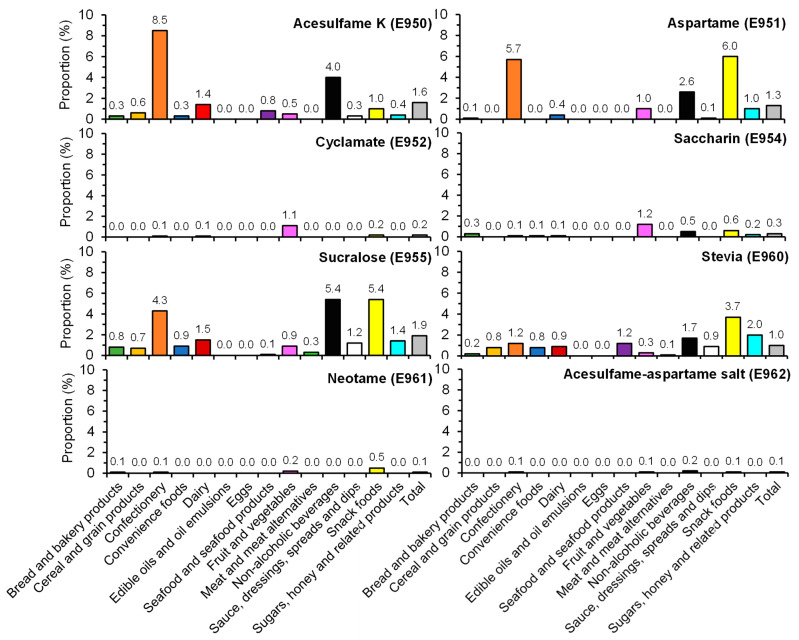
Prevalence of use of different non-nutritive sweeteners (NNSs) by major food category. Data for neohesperidin DC (E959) and neotame (E961) were not presented as only 1 product each was found containing E959 and E961. No product in the dataset contained E956 (alitame) or E957 (thaumatin).

**Figure 2 nutrients-13-01861-f002:**
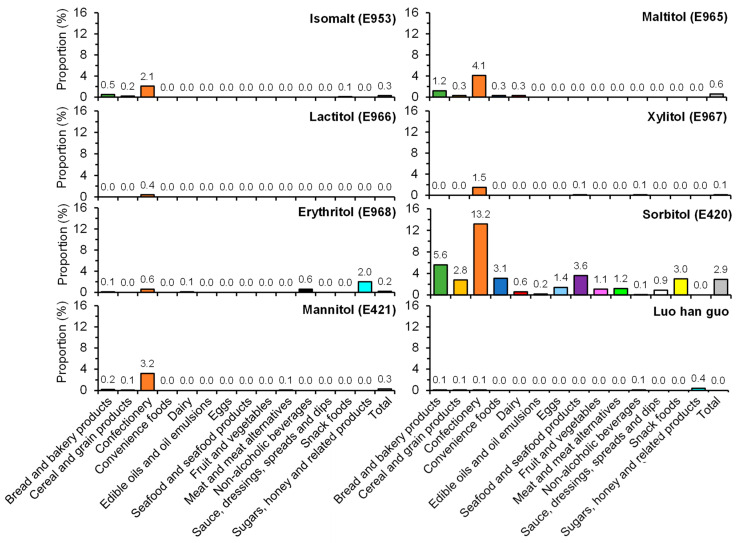
Prevalence of use of different low-calorie sweeteners (LCSs) by food category.

**Figure 3 nutrients-13-01861-f003:**
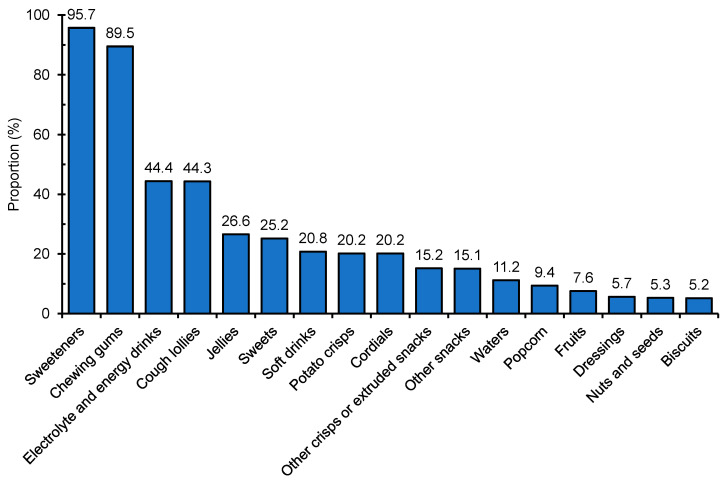
Minor food categories with >5% prevalence of use of non-nutritive (NNSs) and/or low-calorie sweeteners (LCSs).

**Figure 4 nutrients-13-01861-f004:**
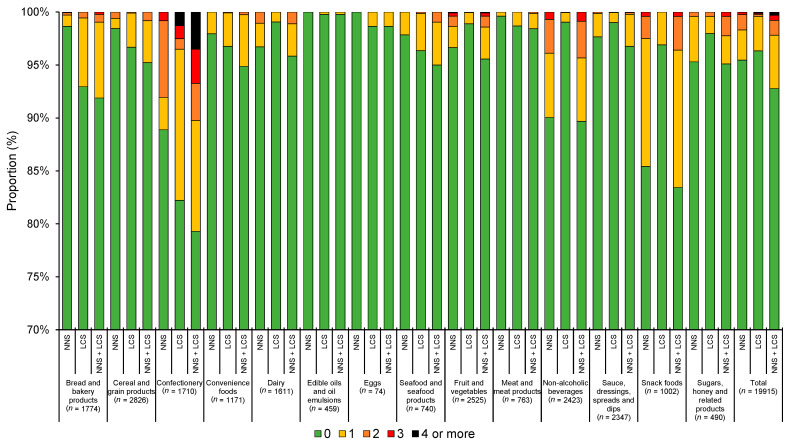
Number of non-nutritive (NNSs) and low-calorie sweeteners (LCSs) used in products by major food category. 0, 1, 2, 3 and 4 refers to the number of NNSs/LCSs found in the food products as stated in the figure caption.

**Figure 5 nutrients-13-01861-f005:**
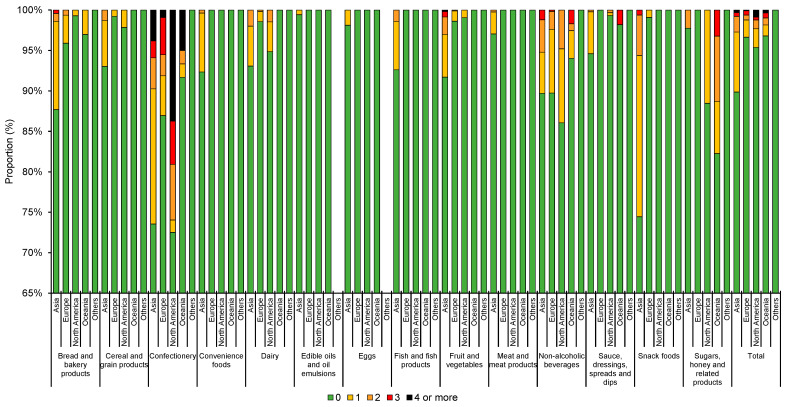
Number of non-nutritive (NNSs) and/or low-calorie sweeteners (LCSs) in food products from different regions by food category. *n* from left to right: 934, 634, 138, 66, 2, 1779, 632, 278, 117, 20, 862, 651, 131, 60, 6, 783, 265, 88, 32, 3, 751, 553, 136, 165, 6, 171, 221, 32, 34, 0, 53, 7, 5, 9, 0, 500, 182, 28, 24, 6, 1157, 866, 420, 37, 45, 406, 262, 68, 20, 7, 1494, 585, 208, 117, 19, 1350, 638, 293, 55, 11, 642, 218, 126, 13, 3, 178, 170, 78, 62, 2, 11,061, 5884, 2029, 811, 130. 0, 1, 2, 3 and 4 refers to the number of NNSs/LCSs found in the food products as stated in the figure caption.

**Table 1 nutrients-13-01861-t001:** Prevalence (*n* (%)) of products with co-presence of non-nutritive sweetener (NNSs), low-calorie sweeteners (LCSs) and free sugar ingredients (FSIs) by food category.

Food Category	Total *n* in Category	NNSs Only	LCSs Only	NNS + FSI	LCSs + FSI	NNSs + LCSs	NNSs + LCSs + FSIs	FSIs Only	No NNSs/LCSs/FSIs
Bread and bakery products	1774	4 (0.2)	18 (1.0)	15 (0.8)	102 (5.7)	0 (0.0)	5 (0.3)	1497 (84.4)	133 (7.5)
*Biscuits*	1085	3 (0.3)	12 (1.1)	13 (1.2)	23 (2.1)	0 (0.0)	5 (0.5)	954 (87.9)	75 (6.9)
*Breads*	181	1 (0.6)	0 (0.0)	0 (0.0)	1 (0.6)	0 (0.0)	0 (0.0)	147 (81.2)	32 (17.7)
*Cakes and other bakery items*	508	0 (0.0)	6 (1.2)	2 (0.4)	78 (15.4)	0 (0.0)	0 (0.0)	396 (78.0)	26 (5.1)
Cereal and grain products	2826	2 (0.1)	4 (0.1)	39 (1.4)	87 (3.1)	2 (0.1)	1 (0.0)	1120 (39.6)	1571 (55.6)
*Breakfast cereals*	466	0 (0.0)	0 (0.0)	0 (0.0)	5 (1.1)	0 (0.0)	0 (0.0)	322 (69.1)	139 (29.8)
*Cereals and nut-based bars*	118	0 (0.0)	0 (0.0)	0 (0.0)	13 (11.0)	0 (0.0)	0 (0.0)	100 (84.7)	5 (4.2)
*Noodles*	1127	2 (0.2)	4 (0.4)	38 (3.4)	68 (6.0)	2 (0.2)	1 (0.1)	566 (50.2)	446 (39.6)
*Pasta, rice or cous cous*	655	0 (0.0)	0 (0.0)	1 (0.2)	1 (0.2)	0 (0.0)	0 (0.0)	67 (10.2)	586 (89.5)
*Other cereal products*	460	0 (0.0)	0 (0.0)	0 (0.0)	0 (0.0)	0 (0.0)	0 (0.0)	65 (14.1)	395 (85.9)
Confectionery	1710	1 (0.1)	7 (0.4)	49 (2.9)	157 (9.2)	79 (4.6)	61 (3.6)	1336 (78.1)	20 (1.2)
*Chocolates*	735	0 (0.0)	2 (0.3)	1 (0.1)	39 (5.3)	14 (1.9)	1 (0.1)	666 (90.6)	12 (1.6)
*Sweets*	658	8 (1.2)	2 (0.3)	15 (2.3)	101 (15.3)	32 (4.9)	16 (2.4)	490 (74.5)	2 (0.3)
*Chewing gums*	57	0 (0.0)	0 (0.0)	0 (0.0)	0 (0.0)	27 (47.4)	24 (42.1)	6 (10.5)	0 (0.0)
*Cough lollies*	61	0 (0.0)	1 (1.6)	2 (3.3)	3 (4.9)	5 (8.2)	16 (26.2)	33 (54.1)	1 (1.6)
*Jellies*	199	1 (0.5)	2 (1.0)	31 (15.6)	14 (7.0)	1 (0.5)	4 (2.0)	141 (70.9)	5 (2.5)
Convenience foods	1171	4 (0.3)	1 (0.1)	18 (1.5)	35 (3.0)	0 (0.0)	2 (0.2)	874 (74.6)	237 (20.2)
Dairy	1611	8 (0.5)	2 (0.1)	44 (2.7)	12 (0.7)	0 (0.0)	1 (0.1)	889 (55.2)	655 (40.7)
Edible oils and oil emulsions	459	0 (0.0)	1 (0.2)	0 (0.0)	0 (0.0)	0 (0.0)	0 (0.0)	6 (1.3)	452 (98.5)
Eggs	74	0 (0.0)	0 (0.0)	0 (0.0)	1 (1.4)	0 (0.0)	0 (0.0)	3 (4.1)	70 (94.6)
Seafood and seafood products	740	0 (0.0)	3 (0.4)	10 (1.4)	18 (2.4)	0 (0.0)	6 (0.8)	274 (37.0)	429 (58.0)
Fruit and vegetables	2525	17 (0.7)	8 (0.3)	67 (2.7)	19 (0.8)	1 (0.0)	0 (0.0)	844 (33.4)	1569 (62.1)
*Fruits*	460	4 (0.9)	0 (0.0)	31 (6.7)	0 (0.0)	0 (0.0)	0 (0.0)	202 (43.9)	223 (48.5)
*Herbs and spices*	586	0 (0.0)	2 (0.3)	3 (0.5)	4 (0.7)	0 (0.0)	0 (0.0)	107 (18.3)	470 (80.2)
*Jam and marmalade*	222	0 (0.0)	6 (2.7)	3 (1.4)	0 (0.0)	0 (0.0)	0 (0.0)	208 (93.7)	5 (2.3)
*Nuts and seeds*	431	6 (1.4)	0 (0.0)	16 (3.7)	1 (0.2)	0 (0.0)	0 (0.0)	83 (19.3)	325 (75.4)
*Vegetables*	826	7 (0.8)	0 (0.0)	14 (1.7)	14 (1.7)	1 (0.1)	0 (0.0)	244 (29.5)	546 (66.1)
Meat and meat alternatives	763	0 (0.0)	0 (0.0)	2 (0.3)	9 (1.2)	0 (0.0)	1 (0.1)	498 (65.3)	253 (33.2)
Non-alcoholic beverages	2423	65 (2.7)	0 (0.0)	162 (6.7)	9 (0.4)	4 (0.2)	10 (0.4)	1332 (55.0)	841 (34.7)
*Coffee, tea and hot chocolates*	915	5 (0.5)	0 (0.0)	29 (3.2)	3 (0.3)	0 (0.0)	0 (0.0)	302 (33.0)	576 (63.0)
*Cordials*	790	2 (2.1)	0 (0.0)	16 (17.0)	1 (1.1)	0 (0.0)	0 (0.0)	74 (78.7)	1 (1.1)
*Electrolyte and energy drinks*	45	4 (8.9)	0 (0.0)	14 (31.1)	0 (0.0)	1 (2.2)	1 (2.2)	25 (55.6)	0 (0.0)
*Fruit and vegetable juices*	527	0 (0.0)	0 (0.0)	15 (2.8)	0 (0.0)	0 (0.0)	0 (0.0)	388 (73.6)	124 (23.5)
*Soft drinks*	672	45 (6.7)	0 (0.0)	81 (12.1)	5 (0.7)	1 (0.1)	8 (1.2)	520 (77.4)	12 (1.8)
*Waters*	170	9 (5.3)	0 (0.0)	7 (4.1)	0 (0.0)	2 (1.2)	1 (0.6)	23 (13.5)	128 (75.3)
Sauce, dressings, spreads and dips	2347	8 (0.3)	5 (0.2)	45 (1.9)	16 (0.7)	0 (0.0)	2 (0.1)	1711 (72.9)	560 (23.9)
*Sauces*	1767	8 (0.5)	5 (0.3)	27 (1.5)	14 (0.8)	0 (0.0)	1 (0.1)	1308 (74.0)	404 (22.9)
*Dressings*	333	0 (0.0)	0 (0.0)	18 (5.4)	1 (0.3)	0 (0.0)	0 (0.0)	218 (65.5)	96 (28.8)
*Spreads and dips*	247	0 (0.0)	0 (0.0)	0 (0.0)	1 (0.4)	0 (0.0)	1 (0.4)	185 (74.9)	60 (24.3)
Snack foods	1001	10 (1.0)	1 (0.1)	125 (12.5)	19 (1.9)	0 (0.0)	11 (1.1)	640 (63.9)	196 (19.6)
*Potato crisps*	312	3 (1.0)	0 (0.0)	60 (19.2)	0 (0.0)	0 (0.0)	0 (0.0)	184 (59.0)	65 (20.8)
*Other crisps or extruded snacks*	419	1 (0.2)	0 (0.0)	49 (11.7)	7 (0.0)	0 (0.0)	3 (0.0)	263 (62.8)	93 (22.2)
*Popcorn*	32	0 (0.0)	0 (0.0)	1 (3.1)	1 (3.1)	0 (0.0)	1 (3.1)	25 (78.1)	4 (12.5)
*Other snacks*	238	2 (0.8)	1 (0.4)	15 (6.3)	11 (4.6)	0 (0.0)	7 (2.9)	168 (70.6)	34 (14.3)
Sugars, honey and related products	490	2 (0.4)	1 (0.2)	12 (2.4)	0 (0.0)	9 (1.8)	0 (0.0)	460 (93.9)	6 (1.2)
*Sweeteners*	23	1 (4.3)	1 (4.3)	11 (47.8)	0 (0.0)	9 (39.1)	0 (0.0)	0 (0.0)	1 (4.3)
*Sugars or honey or syrups*	467	1 (0.2)	0 (0.0)	1 (0.2)	0 (0.0)	0 (0.0)	0 (0.0)	460 (98.5)	5 (1.1)
*Total*	19,915	130 (0.7)	44 (0.2)	604 (3.0)	470 (2.4)	93 (0.5)	98 (0.5)	11,484 (57.7)	6992 (35.1)

FSIs: free sugar ingredients; LCSs: low-calorie sweeteners; NNSs: non-nutritive sweeteners.

## Data Availability

Restrictions apply to the availability of these data. Data was obtained from The George Institute for Global Health under a license. Interested parties should contact Mr Fraser Taylor at ftaylor@georgeinstitute.org.au to discuss access permission.

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
