# Peer review of "The Use of Non-Nutritive and Low-Calorie Sweeteners in 19,915 Local and Imported Pre-Packaged Foods in Hong Kong"

_nutrients, 2021, doi:10.3390/nu13061861_

Round 1

Reviewer 1 Report

The article on "The use of non-nutritive and low-calorie sweeteners in 19,915 local and imported pre-packaged foods in Hong Kong" highlights the need to be aware of sweeteners that are added to pre-packaged foods in Hong Kong. The authors assessed the presence of NNSs, LCSs and FSIs in several food categories and results were presented in tables. My comments to improve the article are:

L30: 249 words - not necessary, also the citation quoted next to the abstract is wrong.

Under Introduction, before L38 - a short sentence to define free sugars e.g. "monosaccharides and disaccharides added to foods by manufacturers, including sugars that are naturally present in honey, syrups, fruit juices" as defined by the FAO.

A short reference to the countries that have imposed sugar tax may be added under this section even though the authors referred to this in L246 under discussion.

Some of the results obtained may be better explained, for example, 3.4. on the use of NNSs/LCSs by region of origin. In addition, the discussion section can include a sentence to properly place in perspective the use of NNSs/LCSs in North America, Europe and Oceania.

The Conclusion need to be revised - perhaps a suggestion to the manufacturers on product formulation, regulations in Hong Kong on the imported foods, more training for the awareness of consumers on the interpreting the food labels and their health implications.

Author Response

The article on "The use of non-nutritive and low-calorie sweeteners in 19,915 local and imported pre-packaged foods in Hong Kong" highlights the need to be aware of sweeteners that are added to pre-packaged foods in Hong Kong. The authors assessed the presence of NNSs, LCSs and FSIs in several food categories and results were presented in tables. My comments to improve the article are:

L30: 249 words - not necessary, also the citation quoted next to the abstract is wrong.

Authors’ response: Thank you, these have been amended.

Under Introduction, before L38 - a short sentence to define free sugars e.g. "monosaccharides and disaccharides added to foods by manufacturers, including sugars that are naturally present in honey, syrups, fruit juices" as defined by the FAO.

Authors’ response: To preserve the flow of text, the suggested text is added in lines 39-40 (line number with Mark Ups turned off).

A short reference to the countries that have imposed sugar tax may be added under this section even though the authors referred to this in L246 under discussion.

Authors’ response: Line 43 has been amended to mention sugar tax.

Some of the results obtained may be better explained, for example, 3.4. on the use of NNSs/LCSs by region of origin. In addition, the discussion section can include a sentence to properly place in perspective the use of NNSs/LCSs in North America, Europe and Oceania.

Authors’ response: Thank you for your suggestion. We have amended the text according to improve the interpretation of results (line 189-199), as well as expanded the relevant section in the discussion (lines 239-247).

The Conclusion need to be revised - perhaps a suggestion to the manufacturers on product formulation, regulations in Hong Kong on the imported foods, more training for the awareness of consumers on the interpreting the food labels and their health implications.

Authors’ response: Thank you for your suggestion. The conclusion has been revised (lines 312-316).

Reviewer 2 Report

In The use of non-nutritive and low-calorie sweeteners in 19,915 local and imported pre-packaged foods in Hong Kong, authors Sing O et al have analyzed nutritional information on a major proportion of Hong Kong’s food supply, where 90% of food is imported. Nutritional information was gathered through a custom database. Food products were curated into specific product categories and frequency of specific NNSs/LCSs use was documented. Frequency of NNSs/LCSs presence in food products was also broken down by region of origin. The release of this manuscript is timely due to the increase in NNSs and LCSs in food products worldwide, along with emerging evidence of risks associated with their consumption. As the authors note from their survey, use of NNSs/LCSs is increasingly common in food products not marketed as diet foods (eg savory snacks and bakery items), and conscientious consumers may have difficulty avoiding these additives. Public health and nutrition researchers require greater awareness of the extent of NNSs/LCSs presence in food products to assess consumer exposure.

Strengths of the manuscript:

  • Thorough investigation into NNSs/LCSs prevalence in food products available to consumers in Hong Kong
  • Highlighted the presence of NNS/LCS in non-diet food products
  • Explores food manufacturer’s practice of combining NNS/LCS in food products, along with their co-occurrence with nutritive sugars (eg sucrose)
  • Provides regional insight into pre-packaged foods manufacturing with respect to NNSs/LCSs

Before the manuscript is accepted for publication: the following comments must be answered.

  • Methods: 1 Data source – It was unclear the extent of nutritional data that was available on the FoodSwitch database before launching the survey of NNSs/LCSs and how much was added by the trained research assistants for the purposes of this study. It is also unclear whether FoodSwitch is a purely crowd-sourced (and therefore uncontrolled) source of information or whether it can only be added to by people with special training and access. Greater detail on this resource would strengthen confidence in the quality of the data.
  • Figure 1: This figure seemed unnecessary as the process for exclusion is straightforward and fully described in the methods.
  • Tables 1 and 2: These should be converted into graphs which readily showcase major findings (eg the presence of NNS in potato crisps, as mentioned in abstract). Tables can instead be made available in the supplement.
  • Supplemental tables 1 and 2: Interesting results on the regional use of NNSs and LCSs in specific food products were mentioned in Results 3.4, but the data were hidden in the supplement. I suggest presenting the data in a different style and moving it to the main manuscript. For example, you could merge the results from NNS and LCS in products and present a single figure, as without the granularity of individual sweeteners information there may not be much of a need to differentiate between the two, for the sake of the figure.
  • Additional figure: Including a summary graphic of specific NNS/LCS frequency out of all sweeteners surveyed would be a great at-a-glance figure for researchers interested in sweeteners. This could also show the top categories the main sweeteners were found in. This would emphasize the importance of studying health impacts of these most common sweeteners.
  • NNS/LCS co-occurrence with FSI: Additional insight into the potential risks of consuming foods with both NNS/LCS and FSI would be appropriate when addressing this combination in the discussion.

Additionally, I suggest making the following minor edits:

  • Table 1: Reformat to match Table 2. Values are too crowded in center of table.
  • Figure 2: Graph is difficult to read. Additional space or marks between food categories to differentiate would help.
  • Table 3: Reformat column titles (eg “category” can all be moved to the second line to keep the word together); left align row titles as in Table 2

Author Response

In The use of non-nutritive and low-calorie sweeteners in 19,915 local and imported pre-packaged foods in Hong Kong, authors Sing O et al have analyzed nutritional information on a major proportion of Hong Kong’s food supply, where 90% of food is imported. Nutritional information was gathered through a custom database. Food products were curated into specific product categories and frequency of specific NNSs/LCSs use was documented. Frequency of NNSs/LCSs presence in food products was also broken down by region of origin. The release of this manuscript is timely due to the increase in NNSs and LCSs in food products worldwide, along with emerging evidence of risks associated with their consumption. As the authors note from their survey, use of NNSs/LCSs is increasingly common in food products not marketed as diet foods (eg savory snacks and bakery items), and conscientious consumers may have difficulty avoiding these additives. Public health and nutrition researchers require greater awareness of the extent of NNSs/LCSs presence in food products to assess consumer exposure.

Methods: 1 Data source – It was unclear the extent of nutritional data that was available on the FoodSwitch database before launching the survey of NNSs/LCSs and how much was added by the trained research assistants for the purposes of this study. It is also unclear whether FoodSwitch is a purely crowd-sourced (and therefore uncontrolled) source of information or whether it can only be added to by people with special training and access. Greater detail on this resource would strengthen confidence in the quality of the data.

Authors’ response: Agree. More details have been added in the methods section (lines 78-80; 93-95; line number with Mark Ups turned off).

Figure 1: This figure seemed unnecessary as the process for exclusion is straightforward and fully described in the methods.

Authors’ response: Agree. Figure 1 now removed.

Tables 1 and 2: These should be converted into graphs which readily showcase major findings (eg the presence of NNS in potato crisps, as mentioned in abstract). Tables can instead be made available in the supplement.

Authors’ response: Agree. Tables 1 and 2 are now provided as supplementary tables 1 and 2. Figures 1 to 3 are provided to highlight the major findings per your suggestions.

Supplemental tables 1 and 2: Interesting results on the regional use of NNSs and LCSs in specific food products were mentioned in Results 3.4, but the data were hidden in the supplement. I suggest presenting the data in a different style and moving it to the main manuscript. For example, you could merge the results from NNS and LCS in products and present a single figure, as without the granularity of individual sweeteners information there may not be much of a need to differentiate between the two, for the sake of the figure.

Authors’ response: Agree. These data are now provided in Figure 5 per your suggestion.

Additional figure: Including a summary graphic of specific NNS/LCS frequency out of all sweeteners surveyed would be a great at-a-glance figure for researchers interested in sweeteners. This could also show the top categories the main sweeteners were found in. This would emphasize the importance of studying health impacts of these most common sweeteners.

Authors’ response: We agree with the reviewer that this information is important and of interest. This information was indeed already provided in Tables 1 and 2 in the original manuscript (now Supplementary Tables 1 and 2). We have however attempted to turn this information into multi-panel figures (Figures 1 and 2) based on our understanding of the reviewer’s request, and are happy to make further improvements to it subject to further comments from the reviewer/editor.

NNS/LCS co-occurrence with FSI: Additional insight into the potential risks of consuming foods with both NNS/LCS and FSI would be appropriate when addressing this combination in the discussion.

Authors’ response: We have added additional discussion on this in lines 276-282.

Table 1: Reformat to match Table 2. Values are too crowded in center of table.

Authors’ response: These are now moved to online supplemental tables 1 and 2 and have been reformatted.

Figure 2: Graph is difficult to read. Additional space or marks between food categories to differentiate would help.

Authors’ response: We have added additional space between categories per your suggestion to improve the readability (Figure 4 in the revised manuscript).

Table 3: Reformat column titles (eg “category” can all be moved to the second line to keep the word together); left align row titles as in Table 2

Authors’ response: Done.